# Diagnosing Coronary Artery Disease on the Basis of Hard Ensemble Voting Optimization

**DOI:** 10.3390/medicina58121745

**Published:** 2022-11-28

**Authors:** Hayder Mohammedqasim, Roa’a Mohammedqasem, Oguz Ata, Eman Ibrahim Alyasin

**Affiliations:** Department of Electrical and Computer Engineering, Institute of Science, Altinbas University, Istanbul 34218, Turkey

**Keywords:** coronary artery disease (CAD), feature selection, machine learning, imbalanced data, ensemble voting classifier, optimization, classification

## Abstract

*Background and Objectives*: Recently, many studies have focused on the early diagnosis of coronary artery disease (CAD), which is one of the leading causes of cardiac-associated death worldwide. The effectiveness of the most important features influencing disease diagnosis determines the performance of machine learning systems that can allow for timely and accurate treatment. We performed a Hybrid ML framework based on hard ensemble voting optimization (HEVO) to classify patients with CAD using the Z-Alizadeh Sani dataset. All categorical features were converted to numerical forms, the synthetic minority oversampling technique (SMOTE) was employed to overcome imbalanced distribution between two classes in the dataset, and then, recursive feature elimination (RFE) with random forest (RF) was used to obtain the best subset of features. *Materials and Methods*: After solving the biased distribution in the CAD data set using the SMOTE method and finding the high correlation features that affected the classification of CAD patients. The performance of the proposed model was evaluated using grid search optimization, and the best hyperparameters were identified for developing four applications, namely, RF, AdaBoost, gradient-boosting, and extra trees based on an HEV classifier. *Results*: Five fold cross-validation experiments with the HEV classifier showed excellent prediction performance results with the 10 best balanced features obtained using SMOTE and feature selection. All evaluation metrics results reached > 98% with the HEV classifier, and the gradient-boosting model was the second best classification model with accuracy = 97% and F1-score = 98%. *Conclusions*: When compared to modern methods, the proposed method perform well in diagnosing coronary artery disease, and therefore, the proposed method can be used by medical personnel for supplementary therapy for timely, accurate, and efficient identification of CAD cases in suspected patients.

## 1. Introduction

In the last few years, the incidence of cardiovascular diseases and deaths in developing countries has increased [1]. According to WHO statistics, the number of deaths is expected to increase in the coming years [2]. Coronary artery disease (CAD) is a prevalent condition that affects the heart and the arteries and is responsible for a large number of deaths. An increase in lipid layers in the coronary arteries leads to constricted blood flow, which leads to a lack of oxygen in the heart muscles. Under severe circumstances, a heart attack caused by a shortage of oxygen might result in death [3]. Prevention is better than cure, and it is possible to mitigate this dangerous disease by following a healthy diet, avoiding smoking and alcohol consumption, and exercising regularly, which can help prevent obesity. People suffering from CAD need an early detection and prediction mechanism to be able to take adequate measures in time [4]. Predicting heart diseases is one of the applications of machine learning (ML). In 1956, the American astronaut Arthur Lee Samuel introduced the term artificial intelligence (AI) when he demonstrated its application for designing games as well as for other applications. ML has since had several real-life applications, such as voice and face recognition, self-driving vehicles, robotics, financial and banking services, and so on. In health care, promising results have been shown for applications such as medical analyses, radiology, and aiding health care professionals in making correct decisions [5]. Data mining is not a single-step process; rather, it comprises several procedures, such as data collection, cleaning, processing, evaluation, and visualization [6]. These processes require powerful algorithms that can extract different patterns and interrelationships that contribute to all lifestyles and industries that use powerful mathematical and statistical processes and ML techniques to obtain the best results and are easy to interpret [7].

Several medical studies have proven that early diagnosis of coronary heart diseases is crucial for saving lives and improving outcomes. Thus, many clinicians and researchers are working to develop methods for early diagnosis and classification of CAD, which include ML and data mining techniques. In recent years, these techniques have been widely used for treatment, classification, and predictive modeling, obtaining improved overall predictive performance [5,6,7].

Features and algorithms can be used to determine the performance of models built using ML techniques. Several feature processing approaches, including feature optimization and feature selection, have been used in contemporary research. Model validation and test performance can be improved by selecting subsets of significant features. Several feature selection methods have been used for CAD detection in the literature.

Arabasadi et al. [8] utilized a combination of a genetic algorithm (GA) and a neural network to enhance the initial weight value. The proposed hypermodel increased classification accuracy by 10%, and accuracy reached 93.85% and 92% specificity. In [9], several ML models and four ensemble learning models were used to classify two CAD data sets, namely, Cleveland and Z-Alizadeh Sani. A GA was applied to determine the best clinical features. Four SVM kernel functions were used in the classification process, achieving 94% accuracy with the Z-Alizadeh Sani dataset and 98% classification accuracy with the Cleveland dataset. Nasarian et al. [10] developed a novel feature selection model called heterogeneous hybrid feature selection (2HFS) with four ML models on the Alizadeh Sani Dataset and three additional well-known CAD datasets. The proposed model achieved a high accuracy of 92% with an XGBoost model. Dwivedi et al. [11] used six different ML models to classify heart disease patients. They used a dataset provided by UCI comprising 13 features with two classes, namely, 120 patients with heart disease and 150 patients without heart disease. To evaluate classification performance, eight metrics were used, and a logistic regression model with accuracy and sensitivity of 85% and 89%, respectively, was applied. Naushad [12] used a CAD dataset comprising 364 patients and 284 healthy individuals. Ensemble ML models were used to classify the CAD patients. Several evaluation metrics, such as accuracy (89%), sensitivity, specificity, and area under the curve (AUC) (96^), were used to evaluate performance. Cüvitoğlu et al. [13] used the same CAD dataset in their research. Principal component analysis (PCA) and a student’s t-test were used to extract optimal features from the dataset. 10-fold cross-validation was applied with six models to classify patients with CADs. All ML models achieved accuracy > 80%; the artificial neural network (ANN) model showed the best performance, and it achieved an AUC of 93%. Ayatollahi et al. [14] used a data set comprising 1324 patient records obtained from the AJA University between March 2016 and 2017. After the dataset was cleaned and arranged, it was divided into training and testing groups. To identify CAD patients, two ML models were applied, namely, SVM and ANN. SVM showed better results than ANN, with 92% sensitivity and 74% specificity. Abdar et al. [15] used the same CAD dataset, which comprised 303 patient records and 54 features. After data preprocessing (normalization and categorical encoding), a new hyper optimization model was used by combining GA and PSO to extract the optimal features and optimize the classifier parameters based on 10 cross-validation experiments; overall, 10 ML models were applied to predict CAD patients. Three models, namely, LinSVM, NuSVM, and C-SVC, were applied as the best models with high accuracies of 93%. Akella et al. [16] used a CAD dataset comprising 303 records and 75 attributes; it was taken from the UCI repository, which contains records since 1988. Six ML models were applied to predict CAD patients. The dataset was split into different sizes as training and test sets. All models achieved accuracy > 80%; the ANN model was selected as the best model with accuracy and sensitivity of 93% and 94%, respectively. Kutrani et al. [17] used the Benghazi Heart Center dataset, which contained 1770 samples and 11 attributes. WEKA software was used to manipulate and evaluate the data. Five ML models were used to classify cardiovascular diseases. The Naïve Bayes model was selected as the best model, with accuracy and true positive (TP) of 88% and 97%, respectively. Tougui et al. [18] used the cardiovascular disease data provided by the UCI. The data set comprised 139 patients and 164 non-patients with 13 attributes. Six ML models and six data mining tools were used in their experiments. Ten cross-validation experiments were used to split the data, and ANN was selected as a good model by using Matlab software with accuracy >86%; also, SVM was selected as a good model with RapidMiner software with specificity >94%. Chen et al. [19] evaluated percentage estimations of the total mean of the research variables, including the first or third quartile of the extra parameter, use ranked set sampling (RSS) and extreme rank group sampling (ERSS). The statistical findings revealed that the RSS and ERSS estimation methods are much more effective than their SRS equivalents.

Previous research has found that the performance of diagnosing coronary artery disease datasets is mostly related to developing different modeling to increase the classification performance [11,12]. The problem of imbalance in medical data sets such as CAD data leads to bias in the ML model during the training process towards the data set with a larger sample size. Furthermore, feature selection is critical, which involves deciding on a subset of features to refer to as classes. The best selection of features is based on two crucial components: To begin, noise must be filtered out and redundant features removed, which can lead to a significant loss in detection accuracy. As a result, the current study suggests combining the hyperized oversampling (SMOTE) approach with the feature selection (RFE) method to increase the performance of the CAD prediction model. This is the first time, to the best of our knowledge, that hard ensemble voting optimization and hyper (oversampling and feature selection approaches) have been employed to increase CAD prediction accuracy. Finally, the outcomes of this study may be utilized to evaluate the performance of an ML model in improving unbalanced datasets and establishing accurate detection methods.

In this study, we developed a novel CAD prediction ML model in which the distribution samples in the data set were obtained using the SMOTE approach to solve the problem of imbalance classes in the data set. Furthermore, we converted the category data into dummy variables and employed the scaling approach for data normalization and the hybrid of two ML methods to create a predictive CAD model. The first method was used to extract features from the dataset, which ultimately helped increase the accuracy and computing time of the newly suggested system. To decrease the dimensions of the data set, the proposed approach used recursive feature elimination (RFE) based on a random forest (RF) model to select the new optimal feature set from the features of the original data set, providing complementary information regarding the CAD predictions. Additionally, we assigned weights to the features depending on their relevance to the target features. The second method was a classification algorithm; in it, the proposed CAD prediction model employed a voting set classifier for classification based on four traditional ML models. To enhance the performance and effectiveness of the voting classifier, we used a network search method to determine the ideal parameters of traditional ML models.

The rest of this paper is organized as follows: Section 2 describes the proposed ML model approach. Section 3 shows the performance of our experimental results compared to contemporary methods. Section 3 discusses the classification performance of the proposed model. In Section 4, we present our conclusions and future work.

## 2. Materials and Methods

In this study, raw data from a CAD dataset were converted to a CSV file and then forwarded to a preprocessing module. In preprocessing, the data were checked for any missing values; if there were no missing values, then data normalization was used to ensure that the data were in the same range. Two main processes, namely, resampling and feature extraction, were combined into one block. These were combined into one block so that they can work in parallell; resampling, the first component, worked to overcome the problems of unbalanced data, whereas the second component, feature extraction, worked as a feature elimination model, as it identified all crucial characteristics from the dataset and reduced data dimensionality. Based on binary class balancing, the results showed a considerable decrease in the false-positive values and an increase in accuracy; this was because ML is sensitive to skewed values. Further, the grid search method was employed to determine the ideal parameters for tuning the ensemble voting classifier with four ML models, which improved the rating performance of the voting classifier. Figure 1 presents the proposed methodology of our work.

### 2.1. Dataset Description

We used the Z-Alizadeh Sani dataset to classify CAD patients [20]. This dataset contains 303 samples (216 CAD patients and 87 normal) with 54 features. The basic features of the data set were demographic, symptoms and examination, laboratory and echo, and ECG. Overall incidence of coronary artery diameter stenosis was considered 50% on coronary CAD angiography, while its absence was considered normal. The classes were labeled as 1 for CAD patients and 0 for normal patients. As shown in Figure 2, the dataset was imbalanced, as the normal class had two times more distribution than the CAD class.

### 2.2. Dataset Standardization

The CAD dataset was subjected to the data normalization procedure. Standardization ensured internal data management, ensuring that all information has the same formatting and substance and making the dataset more relevant [21]. The dataset was modified to have a mean of 0 and a variance of 1.
(1)$$Z=\frac{x-\mu }{\sigma }$$
where X represents the values of each attribute, μ represents the means of each attribute, σ represents the standard deviation of the data set, and Z represents the attributes in a standardized form.

### 2.3. Data Resampling

In terms of data balance, the Z-Alizadeh Sani dataset has 216 and 87 samples with and without CAD, respectively. Thus, the dataset was imbalanced. This imbalance may lead to bias in the results. As shown in previous studies [22,23], if an ML classifier is trained on imbalanced data, the model’s performance is biased as it prefers the majority class while disregarding the minority class. Thus, using a binary class imbalanced CAD data set in the classification model results in a high sensitivity value (where the normal samples (class 0) belong to the majority class) and a reduced specificity value (where the CAD samples (class 1) belong to the minority class). To obtain accurate results, the distribution of samples must be balanced before the classification process. Herein, the synthetic minority oversampling technique (SMOTE) was used for rebalancing data. SMOTE, which is one of the most popular methods, analyzes minority classes and identifies the closest neighbor to every value in the minority class to randomly generate new synthetic samples at a given location; this procedure is repeated until the data samples are balanced and the minority and majority classes are equal. Figure 3 shows the new data distribution after using the SMOTE method; the samples of the minority class (CAD patients) increased from 87 to 216, making the two classes equal.

### 2.4. Feature Selection

Feature selection approaches attempt to eliminate irrelevant features while focusing on those that affect the most reliable target features. Thus, the processing complexity of modeling is reduced, and prediction accuracy is increased by decreasing the number of features. RFE, which is a wrapper feature selection technique that can help select the best classification feature set, maximizes the functionality of the model by removing irrelevant features [24]. In the RFE mechanism, first the complete data set is input into the RFE algorithm, and every feature is evaluated based on the RF classifier. In each loop iteration, RFE utilized the accuracy achieved from the RF model to estimate the importance of one feature and delete features of least importance [25]. In this study, we used RF-RFE to identify the most beneficial features in the CAD data, because the data set contained several non-significant features, and using all of them in the classification stage will reduce predictability and increase time.

### 2.5. Machine Learning

The use of AI tools facilitates the classification and evaluation of the validity of results. In this study, four classifiers were used to effectively classify CAD using the Z-Alizadeh Sani dataset, namely, the RF, AdaBoost, extra tree, and gradient-boosting classifiers. All ML models were integrated to evaluate the ensemble voting classifier.

#### 2.5.1. Random Forest

An RF model comprises several decision trees. It averages the expected results of trees, creating a forest. The method also incorporates three random options for randomly picking training data. In trees, nodes are divided by randomly picking specific categories of attributes and considering only a sampling of all variables for every bare decision tree. Each basic tree trains from a random sample of the dataset throughout the RF training stage [25].

#### 2.5.2. AdaBoost

In the boosting method, weak training is changed into powerful training. AdaBoost, a current type of boosting method, is an array structure used to improve the predictions of each strategy instruction. The goal of boosting is to successfully teach weak classifiers to change previous predictions. In this concept, the model first attempts to fit a prediction on the basic dataset and then fits multiple versions of the exact model on the same dataset. The weights of the items are altered during the training procedure based on the most recent prediction inaccuracy; thus, the resultant model concentrates on tough things [26].

#### 2.5.3. Extra Tree

An extra tree learns from a parental collection by segmenting major data into numerous subgroups (child samples) and predicting separately from each subgroup. It derives the final prediction from the sum of all subgroup predictions. It uses averaging to enhance prediction and cope with overfitting at the same time. The extra tree approach is distinct from other tree-based ensembles as it picks random split points from separate subgroups, can randomly separate subgroups, and can build trees with the whole learning sample [27].

#### 2.5.4. Gradient Boost Classifier (GBC)

GBCs are groups of algorithms that integrate several learning sub-models to provide an accurate prediction. Decision trees are typically used to increase the gradient. Gradient-boosting is a popular ML strategy for addressing regression and classification issues; it can generate a predictor from a group of poor-quality models. Gradient trees, which have poor learning capabilities, usually outperform RFs. Unlike earlier techniques, the GBC model is applied by reducing an independently differentiable loss function.

#### 2.5.5. Voting Classifier

The right selection from a group of several possibilities is obtained using a voting mechanism. Thus, many categories may be selected from various options [28]. The choices of the majority are considered to form the ultimate decision. When many algorithms work on a similar problem, a superior solution might be obtained. By employing ensembles over many classes, everybody cannot make the same mistake.

### 2.6. Evaluation Metrics

In this study, four performance measurements were used to evaluate the proposed ML model on the Z-Alizadeh Sani dataset: accuracy, precision, recall, and F1 score, as indicated in Equations (2)–(5).
(2)$$\mathrm{Accuracy}=\frac{\mathrm{TP}+\mathrm{TN}}{\mathrm{TP}\;+\;\mathrm{TN}\;+\;\mathrm{FP}+\;\mathrm{FN}}$$
(3)$$\mathrm{Precision}=\frac{\mathrm{TP}}{\mathrm{TP}\;+\;\mathrm{FP}}$$
(4)$$\mathrm{Recall}=\frac{\mathrm{TP}}{\mathrm{TP}+\;\mathrm{FN}}$$
(5)$$F1\;\mathrm{score}=\frac{2\cdot \mathrm{TP}}{2\cdot \mathrm{TP}\;+\;\mathrm{FP}+\;\mathrm{FN}}$$
where TP (True Positive), TN (True Negative), FN (False Negative), and FP (False Positive) denote the numbers of correctly classified CAD samples, correctly classified non-CAD samples, wrongly classified CAD samples assorted as not CAD, and incorrectly assorted CAD samples assorted as CAD, respectively [21].

The AUC, along with other metrics, is a crucial performance indicator for assessing ML models. AUC, along with other measures, is often used as a grading scale to assess the performance of classification algorithms on imbalanced data. AUC used the parameters True Positive Rate (TPR) and False Positive Rate (FPR) as indicated in Equations (6) and (7).
(6)$$\mathrm{TPR}=\frac{\mathrm{TP}}{\mathrm{TP}\;+\;\mathrm{FN}}$$
(7)$$\mathrm{FPR}=\frac{\mathrm{FP}}{\mathrm{FP}+\mathrm{TN}}$$

### 2.7. Hyperparameters of ML Models

Each ML classifying model requires one or more parameters to control (efficiency) the prediction results of the classifier. Selecting the appropriate parameter values is challenging and requires a mutual search between the generality of the models and their complexities [19]. Based on a different set of parameter estimates, the grid search value changes before the parameter values are determined. Furthermore, each parameter is increased at specific, validated intervals. To optimize the performance of the models considered in the study, a grid search was conducted through a grid of parameters to determine the best performance parameters. Moreover, optimization was used to reduce prediction time, overprocessing, and error rate, as well as to locate the ideal hyperparameters.

## 3. Results

The results acquired using the Z-Alizadeh Sani dataset specified in Section 2.1 are presented in this section. The Z-Alizadeh Sani dataset has two general problems: high-dimensionality and imbalanced datasets; these issues can lead to inaccurate diagnosis and impact performance outcomes. Thus, a novel ML model is used herein to address the challenges of the Z-Alizadeh Sani dataset. First, during preprocessing, the label encoder technique from the sklearn package was used to transform 22 characteristics into numerical representations; then, using the standard scaler method, data standardization was achieved to ensure uniform distribution of all data; finally, SMOTE was used to overcome the problem of unbalanced data. In feature selection, the 10 best features were found in the Z-Alizadeh Sani dataset based on the RF-RFE accuracy score. To evaluate the performance of our models and compare them with previous models, four ML models and hard ensemble voting optimization (HEVO) were employed to classify the Z-Alizadeh Sani data set. To validate the data, the 70–30 test-train retest split method and then 5-fold cross-validation were used. 70% of the data was used for training (302 samples), and the remaining 30% was used for testing (130 samples). To evaluate the prediction results of each proposed model, we computed its accuracy, precision, recall, and F1 score. Table 1 summarizes the results of all ML models after preprocessing using the split-test training method. The ensemble voting and gradient boosting models produced the best results, with accuracies of 97.5% and 97%, respectively.

The training data (302 samples) obtained from the train-and-test split method was used as input in the 5-fold cross-validation method. Table 2 shows the evaluation results with the mean ± standard deviation of our models after applying the preprocessing techniques with five-fold cross-validation experiments. The optimal evaluation performance was achieved using gradient-boosting, with the highest accuracy and precision of 96% and 97%, respectively.

In Table 3, we used grid search to fine-tune the parameters for all classifiers using the five-fold cross-validation method.

As shown in Table 4, the evaluation results (at least 96.00%) showed the best performance with all ML models. The RF, AdaBoost, gradient-boosting, and extra tree models were developed and optimized based on the HEV classifier, and the classification performance achieved an accuracy of 97.1% and a precision of 98%.

A confusion matrix was used to analyze all ML performance models using the 10 best features obtained using the RF-RFE method. Figure 4 shows the confusion matrix plotted for our five models for the classification of CAD and non-CAD patients. The plot shows the data with true and predicted labels. The diagonal displays the TP and TN instances, showing that the classification is accurate; the testing data utilized for all methods were the same.

The feature importance mechanism provides a rating for features based on how accurately they predict a target variable. Figure 5 shows the most important features after applying the feature selection method based on RF classification; chest pain and age typically have the most considerable influence in classifying whether patients have CAD.

ROC or AUC analysis is one of the most accurate measures for evaluating the efficiency of an imbalanced dataset; a desirable ratio is between 80% and 90%, and >90% is usually better. In Figure 6, the ROC curve versus the TP and FP ratios is displayed.

As shown in Table 4, we evaluated the performance of our proposed system based on the results of previous studies that used the Z-Alizadeh dataset. According to the previous studies, none of them have used the hard ensemble voting optimization model based on hyper oversampling and the feature selection preprocessing model for classifying CAD data. In this study, our proposed model outperforms previous studies in terms of binary-classification problems, achieving an accuracy of up to 97% with five-fold cross validation. Furthermore, the RF-RFE approach was used to select 10 ideal features using our proposed model, which appeared to be the most important for the CAD prediction method. This is the smallest number of features identified in contemporary literature. Many of the missing results in Table 5 were marked as MR; thus, it is implied that some outcome indicators are not recorded in the literature. The performance of medical applications must be assessed using matrices such as AUC and F1-score, specifically with imbalanced datasets. The column “NF” in Table 4 indicates the optimal number of features obtained for training and testing the models based on different feature selection methods. Some abbreviations used in the table include accuracy (ACC), precision result (Pr), and recall result (Rr). The RF-RFE approach was used to select 10 ideal features using our proposed model, which appeared to be the most important for the CAD prediction method. This is the smallest number of features identified in contemporary literature.

## 4. Discussion

To achieve the optimal performance for CAD diagnosis, we have proposed a new ML model for CAD that includes data preparation, oversampling, feature method, vote classification, and hyper-tuning parameters. After data preparation, SOMTE was used as an over-sampling method to balance the data, as shown in Figure 3. A RF classifier based on RFE was used as a feature selection method. Whereas, feature selection uses a number of engineering methodologies to select the most important features in the prediction process by reducing unnecessary features and noise present in the raw data, which greatly improves the effectiveness of machine classifier training. As shown in Figure 5, the optimal features for CAD classification were reduced to the top 10 using the RFE method compared to the literature. To evaluate the approach to diagnosing CAD rapidly and accurately, the over-all performance of four ML models and hard ensemble voting were tested on a CAD dataset processed by the feature processing method and the dataset oversampling method. As well, five cross-validation methods are used to evaluate the robustness and performance of all models. Grid search was used to find the optimal parameters for four ML models based on a hard voting classifier. The experimental results indicate that the voting set classifier has the best classification results on the CAD data set. 

## 5. Conclusions

Cardiovascular diseases are the leading cause of death worldwide. Several researchers worldwide are attempting to develop methods for rapid and precise CAD identification. Several dataset properties are related to different degrees of CAD. Therefore, to obtain the best classification results, it is crucial to identify the features that affect the classification of CAD. A novel ML model based on HEVO was used in this study to improve the performance of contemporary classification methods for CAD identification on the Z-Alizadeh Sani dataset. Five-fold cross-validation and 70–20 train-test splitting methods were applied with four ML models and the ensemble voting classifier to train the CAD identification model. SMOTE was employed to reconstruct the imbalanced data set by producing new samples and appropriately balancing them. RFE was used to choose and decrease the dimensionality of the data; the 10 best features were determined from 54 features, which is the least number of features reached compared to that in contemporary literature. To enhance the performance of our models, four ML models were optimized based on the HEV classifier. Five evaluation metrics, namely accuracy, precision, recall, F1-score, and AUC, were used to evaluate all our classification models. The highest classification result was reached with a five-fold HEV classifier with accuracy, precision, F1-score, and AUC of 97%, 98%, 97, and 98, respectively. The findings of this study reveal high performance compared to contemporary approaches, indicating that our proposed model is highly competitive and can be used for diagnosis by healthcare experts. In the future, we aim to use Internet of Things methods with bigger datasets.

## Figures and Tables

**Figure 1 medicina-58-01745-f001:**
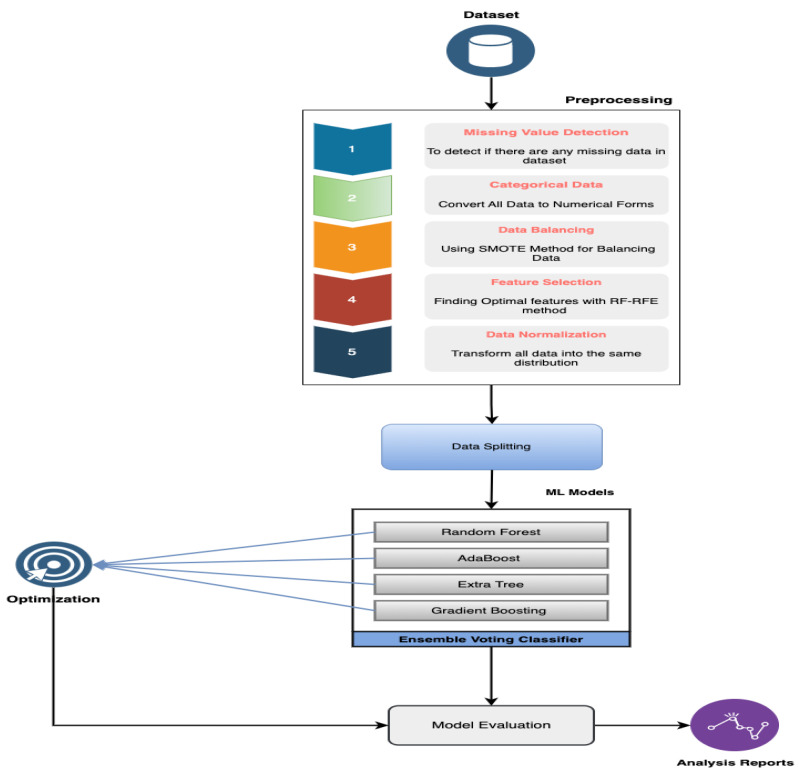
Flowchart for our proposed model.

**Figure 2 medicina-58-01745-f002:**
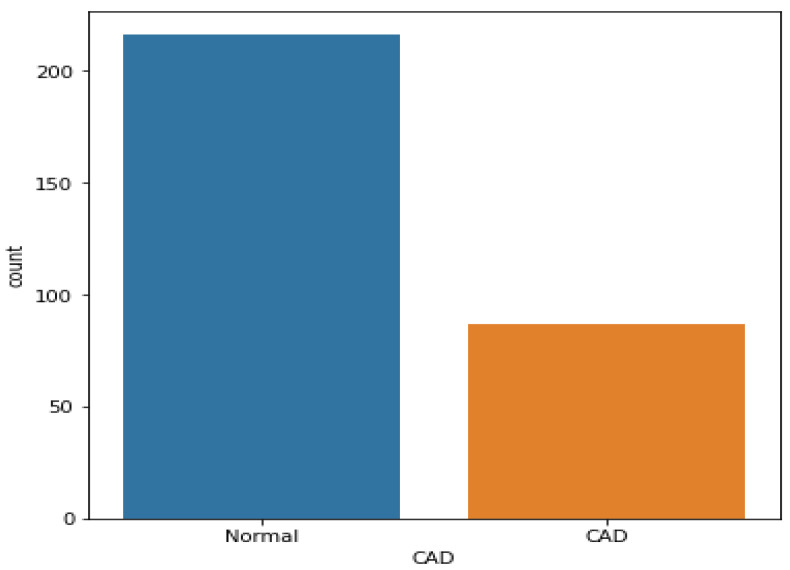
Sample distribution of the Z-Alizadeh Sani dataset.

**Figure 3 medicina-58-01745-f003:**
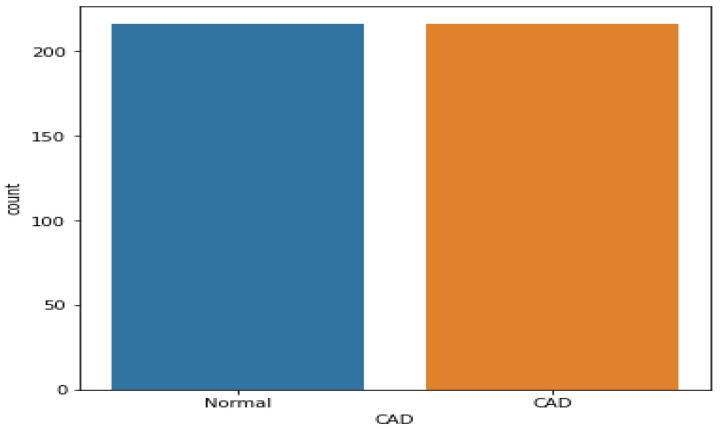
Data balancing using SMOTE.

**Figure 4 medicina-58-01745-f004:**
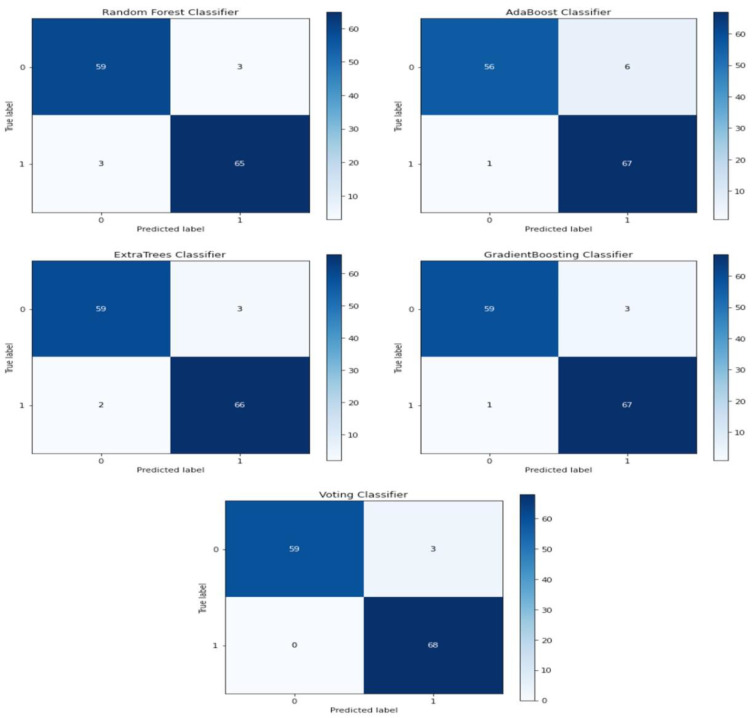
Confusion matrix with all proposed ML models.

**Figure 5 medicina-58-01745-f005:**
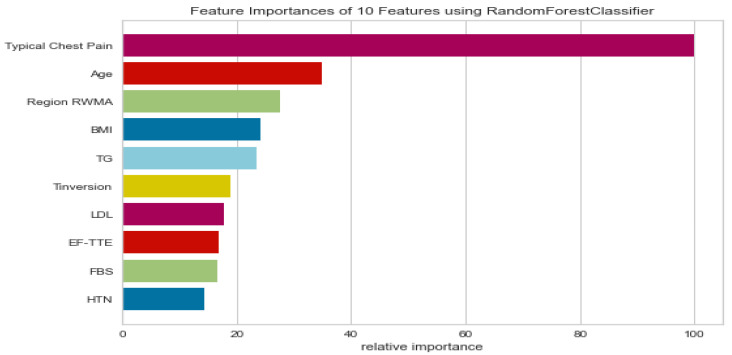
Feature importance using the RF-RFE model.

**Figure 6 medicina-58-01745-f006:**
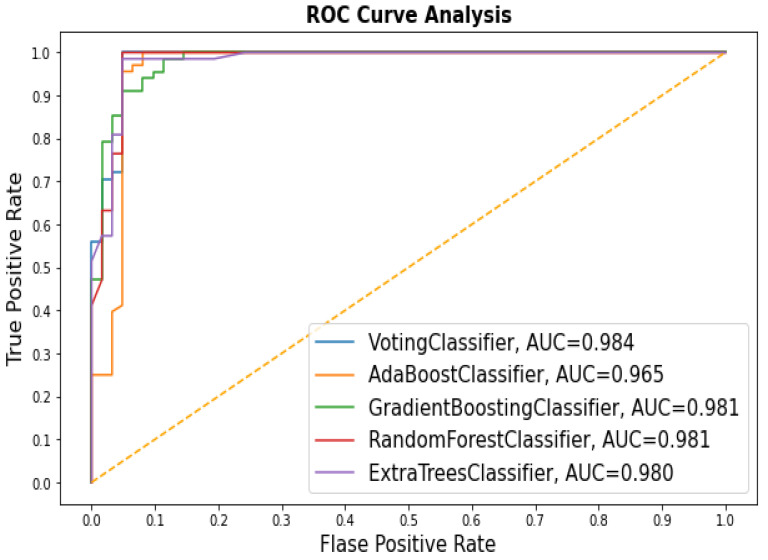
ROC curves of all methods used in the Z-Alizadeh Sani dataset with two classes, CAD and non-CAD. The blue, orange, green, red, and purple colors refer to the developed ensemble voting model with the overall highest score of 98.4%, the AdaBoost model with a 96.5% score, a gradient-boosting model with a 98.1% score, the RF model with an overall highest score of 98.4%, and the extra trees model with a 98% score, respectively.

**Table 1 medicina-58-01745-t001:** Results of all machine learning models using the split-test training method.

Classifier	Accuracy	Precision	Recall	F1-Score
RF	96%	96%	96%	96%
AdaBoost	95%	95%	95%	95%
Gradient-boosting	97%	97%	97%	97%
Extra trees	96%	96%	96%	96%
Ensemble voting	97.5%	97.6%	97.5%	97.3%

**Table 2 medicina-58-01745-t002:** Results of all machine learning models using the five fold cross-validation method.

Classifier	Accuracy	Precision	Recall	F1-Score
RF	94.8% ± 0.07	98% ± 0.05	93% ± 0.08	96% ± 0.06
AdaBoost	95.5% ± 0.09	96% ± 0.09	95.4% ± 0.10	96% ± 0.09
Gradient-boosting	96% ± 0.08	97% ± 0.09	97% ± 0.06	96% ± 0.07
Extra trees	95.3% ± 0.05	98% ± 0.05	94% ± 0.07	96% ± 0.04
Ensemble voting	96% ± 0.06	99% ± 0.05	95% ± 0.08	97% ± 0.05

**Table 3 medicina-58-01745-t003:** Optimal hyperparameters using grid search optimization.

Classifier	Hyperparameters
RF	Criterion = gini, number of estimators = 100, max_depth = 10
AdaBoost	Estimators = 150, learning-rate = 0.1
Gradient-boosting	Estimators = 110, learning-rate = 0.3
Extra trees	Criterion = gini, number of estimators = 100, max_depth = 10

**Table 4 medicina-58-01745-t004:** Experimental results with tuned model hyperparameters with five-fold method.

Classifier	Accuracy	Precision	Recall	F1-Score
RF	96.4% ± 0.07	98% ± 0.09	97% ± 0.06	97% ± 0.06
AdaBoost	96.8% ± 0.07	98% ± 0.09	97% ± 0.06	96% ± 0.06
Gradient-boosting	97% ± 0.05	98% ± 0.05	98% ± 0.06	98% ± 0.05
Extra trees	96.9% ± 0.04	98% ± 0.05	95% ± 0.05	97% ± 0.04
Ensemble voting	97.1% ± 0.07	98% ± 0.09	98% ± 0.06	97% ± 0.06

**Table 5 medicina-58-01745-t005:** Comparison of our proposed model with other contemporary performance models on the Z-Alizadeh Sani dataset.

Study	NF	ACC	Pr	Rr	F1	AUC	Year
[13]	25	86.49	MR	73.61	75	83	2018
[15]	29	93.08	MR	MR	91.51	MR	2019
[10]	16	94.66	94.70	94.70	94.70	96.6	2019
[11]	27	92.58	92.59	92.99	90.62	MR	2020
[29]	22	88.34	92.37	91.85	92.12	MR	2020
[30]	56	MR	75.5	MR	85	86.1	2022
[22]	12	94.2	94.4	94.6	94.3	97	2022
Proposed	10	97	98	98	97	98	-

## Data Availability

Data are contained within the article.

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
