# Peer review of "Diagnosing Coronary Artery Disease on the Basis of Hard Ensemble Voting Optimization"

_medicina, 2022, doi:10.3390/medicina58121745_

Round 1

Reviewer 1 Report

This paper proposed ML framework to classify patients with CAD. For it to be accepted, I would suggest the authors provide a detailed discussion of its novelty compared with prior methods, a clearer description of the training/testing data used, and their evaluation process to address any concerns on the level of evidence presented in the results. 

Introduction

1. Limitations of prior methods should be reviewed. Also, the proposed method seems to use a set of classical ML algorithms to classify CAD patients. It is unclear what novelty the proposed method has compared with prior work. 

Methods

1. What is Cath?

2. Line 168, "More distribution" -> more frequent?

3. Section 2.2, the authors only normalized the mean and variance of the data. However, this does not guarantee that "all information has the same distribution" as they claimed since the data distribution could be different and/or non-Gaussian. I would like the authors to state and justify their assumptions about the distributions of their data. 

4. Similarly, in section 2.3, the authors applied over-sampling in the CAD class. However, this only accounts for the balance of frequencies between two classes, but not necessarily the distributions. 

5. Important information is missing in the dataset description. Only training data is mentioned in this section. The authors should state their training, validation, and testing data and splits. 

Results

1. The authors mentioned "five cross validation", should it be "five-fold cross-validation"? Also, the "train-test" split was also mentioned - which testing method did the authors follow during the evaluation?

2. What data did the authors use to tune the hyperparameters? Was it the validation split? Since there doesn't seem to be a held-out test dataset, tuning the hyperparameters on the validation data where the evaluations were performed will lead to overfitting and deceptive high accuracy in the reported results. 

Reviewer 2 Report

see the attachment 

Round 2

Reviewer 1 Report

The approach of data splitting described in the revised paper to train, tune, and evaluate the ML models does not seem to be correct. I cannot accept this manuscript until this can be addressed.

Table 3 and 4: You cannot tune model parameters using the test set when doing a train/test split. Tuning model parameters on the test split will only provide how the models perform on this test split, but not an estimate of model performance on unseen data. The models could overfit to your test split. A validation set is required in this case where you tune the hyper parameters on the validation set and then test once on a heldout test set. 

I do not see why hyperparameters were tuned twice using both cross validation and train test split. The correct approach of using both cross validation and train/test split should be first splitting the data into train and test sets, and then further splitting the training data to perform cross validation. You could then report your cross validation results, select the best set of model hyperparameters, and report results on the heldout test set. 

Standard deviations should be reported for 5 fold cross validations to demonstrate robustness of models regarding data splits.

There are many grammar mistakes in the Result section. The manuscript requires proofreading for better readability. 
